# Standardized Digital Image Analysis of PD-L1 Expression in Head and Neck Squamous Cell Carcinoma Reveals Intra- and Inter-Sample Heterogeneity with Therapeutic Implications

**DOI:** 10.3390/cancers16112103

**Published:** 2024-05-31

**Authors:** Eric Deuss, Cornelius Kürten, Lara Fehr, Laura Kahl, Stefanie Zimmer, Julian Künzel, Roland H. Stauber, Stephan Lang, Timon Hussain, Sven Brandau

**Affiliations:** 1Department of Otorhinolaryngology Head and Neck Surgery, University Hospital Essen, 45147 Essen, Germanytimon.hussain@uk-essen.de (T.H.); sven.brandau@uk-essen.de (S.B.); 2Department of Otorhinolaryngology Head and Neck Surgery, Molecular and Cellular Oncology, University Medical Center Mainz, 55131 Mainz, Germany; 3Institute of Pathology, University Medical Center Mainz, 55131 Mainz, Germany; 4Department of Otorhinolaryngology, Head and Neck Surgery, University Hospital Regensburg, 93053 Regensburg, Germany; 5Institute for Biotechnology, Shanxi University, No. 92 Wucheng Road, Taiyuan 030006, China; 6Department of Otorhinolaryngology, Klinikum Rechts der Isar, Technical University of Munich, 81675 Munich, Germany

**Keywords:** programmed cell death 1 Ligand 1 (PD-L1), combined positivity score (CPS), tumor proportion score (TPS), heterogeneity, head and neck squamous cell carcinoma (HNSCC), biomarker, prognosis, oropharyngeal squamous cell carcinoma (OPSCC), p16, HPV

## Abstract

**Simple Summary:**

PD-L1 expression determines patients’ eligibility for immunotherapy. Current sampling does not consider the heterogeneity of PD-L1 expression in head and neck squamous cell carcinoma (HNSCC) within the primary tumor. Moreover, potential differences are not considered when comparing primary tumors and their associated metastases or local recurrences, hereby excluding potential responders to immunotherapy. Here, we investigated the inter-sample heterogeneity of PD-L1 expression by analyzing multiple samples from individual patients. Multisection staining revealed clinically relevant CPS changes, which would have potentially affected treatment decisions in 28.7% (intra-tumoral), 44.4% (tumor vs. metastases), and 61.5% (initial tumor vs. local recurrence) of patients, respectively, compared with single-region staining. Increased CPS in primary tumors and lymph node metastases were associated with improved 5-year overall survival. Our results suggest that multiple tumor sections should be evaluated in HNSCC patients when assessing PD-L1 expression prior to potential immunotherapy, particularly if the initial result was negative.

**Abstract:**

For practical reasons, in many studies PD-L1 expression is measured by combined positive score (CPS) from a single tumor sample. This does not reflect the heterogeneity of PD-L1 expression in head and neck squamous cell carcinoma (HNSCC). We investigated the extent and relevance of PD-L1 expression heterogeneity in HNSCC analyzing primary tumors and recurrences (LRs), as well as metastases. Tumor tissue from 200 HNSCC patients was immunohistochemically stained for PD-L1 and analyzed using image-analysis software QuPath v3.4 with multiple specimens per patient. CPS was ≥20 in 25.6% of primary tumors. Intra-tumoral heterogeneity led to a therapeutically relevant underestimation of PD-L1 expression in 28.7% of patients, when only one specimen per patient was analyzed. Inter-tumoral differences in PD-L1 expression between primary tumors and lymph node metastasis (LNM) or LR occurred in 44.4% and 61.5% (CPS) and in 40.6% and 50% of cases (TPS). Overall survival was increased in patients with CPS ≥ 1 vs. CPS < 1 in primary tumors and LNM (hazard ratio: 0.46 and 0.35; *p* < 0.005); CPS in LR was not prognostic. Our analysis shows clinically relevant intra- and inter-sample heterogeneity of PD-L1 expression in HNSCC. To account for heterogeneity and improve patient selection for immunotherapy, multiple sample analyses should be performed, particularly in patients with CPS/TPS < 1.

## 1. Introduction

Squamous cell carcinoma of the head and neck (HNSCC) is the sixth most common tumor type worldwide with over 880,000 new cases and over 400,000 deaths each year [1,2]. Even with extensive treatment using surgery and adjuvant radio-(chemo)therapy, average overall survival across all tumor stages is only about 65% [3,4]. The substantial number of treatment failures and recurrences keeps driving the search for alternative treatment approaches. Heterogenous tumor origination depending on the tumor biology and location, complicate the quest for additional therapeutics. For HPV-associated oropharyngeal squamous cell carcinoma (OPSCC), vaccination is the most promising preventive approach to reduce its incidence, and programs have recently been established in certain parts of the world. However, incidence rates are expected to rise substantially before their effects become apparent in the coming decades [5].

In the search for novel treatment approaches, immunotherapeutic agents have been the latest drugs introduced into the standard treatment protocols for HNSCC in recent years. Anti-PD-1 antibodies nivolumab and pembrolizumab replaced the previously administered cisplatin-based chemotherapy with anti-EGFR antibody cetuximab, known as the EXTREME regimen, as first-line palliative therapy of unresectable relapsed or metastatic RM-HNSCC in the European Union [6,7]. Unfortunately, HNSCC patients across all sub-entities have low objective response rates to these agents of approximately 20%, which is substantially lower than those of melanoma or non-small cell lung carcinoma patients [8,9]. Other immune checkpoint inhibitors such as cytotoxic T-lymphocyte associated protein 4 (CTLA-4), lymphocyte activating 3 (LAG-3), T-cell immunoglobulin and mucin-domain containing-3 (TIM 3), and T-cell immunoreceptor with Ig and ITIM domains (TIGIT) are being evaluated in ongoing palliative clinical trials [10,11].

While immunotherapy can induce long-lasting disease control in a small subgroup of patients that respond, the low overall response rates emphasize the need for predictive and prognostic biomarkers. Previous studies have revealed the predictive value of PD-L1 expression measured by combined positive score (CPS) or tumor proportion score (TPS) in tumor samples [6,12].

Accordingly, determining the level of PD-L1 expression is mandatory prior to the application of pembrolizumab except for in Japan [13]. Therefore, the preference for first-line therapy in RM-HNSCC is currently based on the CPS. In case of a RM-HNSCC with a CPS ≥ 1 < 20, depending on the urgency of a treatment response, combination of pembrolizumab with chemotherapy is the treatment of choice. However, if a CPS ≥ 20 is present, monotherapy with pembrolizumab can be employed. But, in patients with a CPS < 1, chemotherapy-only regimens like TPEX (docetaxel, cisplatin, cetuximab) or EXTREME (5-fluorurcacil, cisplatin, cetuximab) are still the standard-of-care. In case of platinum-refractory first-line therapy without a concurrent immunotherapeutic agent, not CPS but TPS ≥ 50 is required as a biomarker before initiation of second-line immunotherapy with pembrolizumab [6].

Interestingly, some HNSCC patients with a CPS < 1 have been shown to respond well to pembrolizumab while others with CPS ≥ 20 showed no anti-PD-1 treatment response. Also, treatment responses may substantially differ in primary tumors compared to lymph node metastases, as shown in neoadjuvant studies [11]. The variability has been ascribed to multiple influencing parameters related to the heterogeneity of the tumor microenvironment. These include the diversity of the T-cell receptor repertoire, the number of tumor-infiltrating T lymphocytes, and the tumor mutational burden. Most obviously, heterogenous PD-L1 expression within the tumor influences the response to anti-PD-1 treatment and may be a major confounding factor. When determining the degree of expression and planning therapy, CPS is typically measured in a single tissue sample, therefore not taking into account potential intra-tumoral variability [14]. Also, scoring systems, staining protocols, and criteria for positivity differ between institutions in previous studies.

In this study, we measured the degree of intra- and inter-tumoral PD-L1 expression heterogeneity, while reducing technical and clinical confounding factors. We tested the intra- and inter-tumoral concordance of CPS and TPS based on PD-L1 staining within two samples from the same tumor specimen or primary tumor versus related metastasis and recurrences. We also compared PD-L1 expression in primary tumors, sites of local recurrence, lymph nodes, and distant metastases within the same patient to assess intra-patient inter-tumor variability. Furthermore, we evaluated clinicopathologic, predictive, and prognostic associations.

## 2. Materials and Methods

### 2.1. Patients and Samples

A total of 310 tissue samples obtained from 200 patients with primary or local recurrent HNSCC, diagnosed between January 2010 and January 2015, were included in the analysis. Specifically, the tissue samples analyzed consisted of 198 primary tumor samples from 168 patients, 36 locally recurrent tumor samples from 27 patients, 67 lymph node metastasis samples from 54 patients, and 9 distant metastasis samples from 8 patients (Figure 1, Table 1). Exclusion criteria were a history of other malignancies, as well as missing biobank tumor tissue and relevant clinical data. Clinically collected data from patients’ files included gender, tobacco/alcohol consumption, tumor location, tumor stage, age at first diagnosis, last contact, date of death, and treatment modality. Patients were staged equally using the TNM system from the seventh Edition of UICC 2010. The classification system was not adapted to the eighth TNM classification for malignant tumors for better comparability. In addition, pathologic data was retrieved. This included lymph node ratio, extranodal extension, residual status, grading, and p16^4INKa^ expression status. Survival data were ascertained with the aid of the Cancer Registry Rhineland Palatinate. All experiments were performed in accordance with relevant laws and ethical guidelines. This study was approved by the ethical review committee of the Medical Association of Rhineland Palatinate (837.485.15 (10253) 29 January 2016) and by the Ethics Committee of the University Medical Center Essen (21-9877-BO; 11 February 2022).

### 2.2. Tissue Microarray Construction

To reduce the probability of staining variability between whole slide sections, tissue micro arrays were prepared from formalin-fixed, paraffin-embedded (FFPE) excess tumor tissue. For this purpose, corresponding hematoxylin and eosin (HE) sections were analyzed for adequate viable tumor cells. Two representative non-neighboring tissue areas were randomly marked with a permanent marker on HE slides for subsequent punching. Then, 1.6 mm tumor punches were manually removed from corresponding areas of the donor block using a tissue microarrayer (TMArrayer Pathology Devices, San Diego, CA, USA) and separately divided into two recipient blocks. Two cores of human muscle tissue were placed in each recipient block to serve as a negative control and a reference for orientation. For later identification, samples’ positions were marked in a grid [15].

### 2.3. Immunohistochemical Staining

To evaluate the heterogeneity of membranous PD-L1 expression in HNSCC, tumor tissue was fixed at least for 24 h in 10% neutral buffered formalin and embedded in melted paraffin with controlled temperature under 60 °C. TMA sections (4 µm thickness) were cut with a microtome and mounted on adhesive slides (Superfrost Plus, ThermoScientific Waltham, MA, USA) at 58 °C for one hour. Deparaffination and rehydration were performed twice in xylol for 10 min, followed by 100%, 100%, 90%, 80%, and 70% solutions of ethanol and twice-distilled water for three minutes each. Samples were immunohistochemically stained with the validated PD-L1 IHC 22C3 pharmDx kit (Agilent Technologies, Santa Clara, CA, USA), which is approved for PD-L1 staining and interpreting of CPS/TPS in HNSCC. Antigen demasking was performed in preheated low pH EnVision FLEX Target Retrieval Solution (ready to use (RTU), Agilent Technologies, Santa Clara, CA, USA) by heat-induced antigen retrieval (HIER) in a steamer at 95 °C–99 °C for 20 min. Afterward, the sections cooled down to 65 °C in target retrieval solution and were put for a further five minutes in Envision FLEX Wash Buffer at room temperature (Agilent Technologies, Santa Clara, CA, USA).

Endogene peroxidase was blocked using DAKO REAL peroxidase blocking solution (RTU, Agilent Technologies, Santa Clara, CA, USA). Samples were incubated with the primary anti-PD-L1 antibody (Clone 22C3; monoclonal IgG1 mouse anti-human antibody; ≈3 µg/mL, RTU, Agilent Technologies, Santa Clara, CA, USA) at room temperature for 30 min. Staining was automatically performed using the Dako Autostainer Link 48 (RTU, Agilent Technologies, Santa Clara, CA, USA). Slides were covered with 200 µL of solutions for preprogrammed times each. Primary antibody binding was detected by incubation with secondary anti-mouse antibody (RTU, EnVision FLEX+ Mouse LINKER) to enhance the signal and afterwards with horseradish–polymer conjugated tertiary antibodies (RTU, Agilent Technologies, Santa Clara, CA, USA) for 30 min each. For visualization, Dako DAB + Chromogen and DAB + substrate buffer (RTU, Agilent Technologies, Santa Clara, CA, USA) was used for 10 min. Subsequently, the color of the precipitated DAB was modified by EnVision FLEX DAB Enhancer (RTU, Agilent Technologies, Santa Clara, CA, USA) for 5 min. Counterstaining was performed by incubating for 5 min with DAKO REAL Hematoxylin (Agilent Technologies, Santa Clara, CA, USA) and bluing in tap water for 3 min. After each step, slides were cleaned with Dako Wash buffer (tris-buffered saline (TBS)) to elute the prior reagents for 5 min. Afterward, dehydration was performed using distilled aqua, ascendant ethanol solutions, and xylol to preserve the staining results. Subsequently, mounting was performed with Enthelan (Merck, Darmstadt, Germany) and coverslips.

### 2.4. Immunohistochemical Scoring

Scoring of PD-L1 in non-small cell lung cancer and urothelial carcinomas by QuPath has already been validated and has shown a good reliability [16,17]. So, we decided to evaluate membranous staining of PD-L1 in an automated, quantitative, objective, and standardized manner by using the analysis software QuPath v3.4 (Open Software for Bioimage Analysis) after scanning the slides with 200× magnification (Nanozoomer HT2.0, Hamamatsu Photonics, Hamamatsu, Japan) [18]. Tissue areas were classified into stroma, tumor, and immune cells (Figure 2). The analysis was not started before manual control of the classified tissue in TMA spots had taken place. Parameters for positive cell detection are summarized in Appendix A. In addition, results were checked after analyses for correctness according to the pharmDX staining manual in HNSCC. One requirement was at least 100 viable tumor cells in each tissue sample. Positive staining was defined as partial or complete membranous staining that is perceived distinct from cytoplasmatic staining. To evaluate staining results of PD-L1 expression, the combined positive score (CPS) and tumor proportion score (TPS) were determined (Figure 3). Indeed, CPS can exceed the value of 100, but the maximum value is limited to 100 by definition [19].

The values were calculated as follows:$$Combined\;Positive\;Score=100\times \frac{Number\;of\;PDL1\;positive\;tumor\;cells}{Total\;number\;of\;viable\;tumor\;cells}$$
$$Tumor\;Proportion\;Score=100\times \frac{\begin{array}{c} Number\;of\;PDL1\;positive\;cells\\ \left(tumorcells,\;lymphocytes,\;macrophages\right) \end{array}}{Total\;number\;of\;viable\;tumor\;cells}$$

### 2.5. Assessment of Heterogeneity and Reclassification

To assess intra-tumor heterogeneity, CPS and TPS were determined on two separated samples from one tumor specimen or biopsy. The calculated PD-L1 expression of the first sample was used as reference value. The absolute PD-L1 expression score value was used to classify the tumors into the following categories: CPS < 1, CPS ≥ 1 < 20, CPS ≥ 20, TPS < 1, TPS ≥ 1 < 50, and TPS ≥ 50. Subsequently, potential deviations between the specific categories of both intra-tumoral samples were determined and specified as a reclassification in percent. For the assessment of inter-tumoral heterogeneity between primary tumor location, primary tumor with recurrent tumors, or its metastasis, first mean values of both intra-tumoral samples were calculated. Then, mean values ware compared with each other, and changes between absolute values and categorical score were measured.

### 2.6. Statistical Analysis

Statistical tests, charts, and images were performed using SPSS (v28.0.0.0, IBM, Armonk, NY, USA), GraphPad Prism (v9.4.1 GraphPad Software Inc., San Diego, CA, USA), and Excel 365 (Microsoft, Redmond, WA, USA). Pearson’s chi-square test for goodness of fit was used to evaluate association und differences between frequencies of categorical data. Correlations for non-parametric ratio datasets were assessed by Spearman’s rho, for ordinary data by Kendall Tau-B and for dichotomous data by Phi. For samples sizes under five, estimation is based on r/z transformation by Fisher. Metric characteristics were assessed using boxplots, histograms, and Shapiro–Wilk test. CPS and TPS showed no normal distribution, so that non-parametric two-tailed unpaired tests like Mann–Whitney U or Kruskal–Wallis test and two-tailed paired tests like the Wilcoxon und Friedman tests were used for estimating differences between the groups. Correction for multiple testing was performed according to Dunn. Differences in 5 year overall survival based on categorized CPS and TPS in primary tumors and local recurrences were visualized using Kaplan–Meyer curves and compared by log rank test. By Cox regression, the prognostic value of CPS and TPS was calculated after adjusting for multivariate established clinicopathological factors like tumor staging and p16-status. A *p*-value of ≤0.05 was considered significant. Overall survival was defined as the period between the date of diagnosis and the date of death or last contact (censored). Due to a small sample size (n = 29) of primary tumors treated by radio(chemo-)therapy, a possible predictive value of CPS and TPS was not evaluated by logistic regression but by correlation of bicategorical variables, approximately. Treatment response was assessed according to RECIST 1.1 criteria. Complete and partial responses were considered as response; stable or progressive disease was considered as no response.

## 3. Results

### 3.1. Patients

General patient details are displayed in Table 2.

OPSCC was the most common type of primary HNSCC in the study and accounted for 63.7% (107/168) of patients. The proportion of p16-positive OPSCC was 32.7% (35/107). Primary tumors were diagnosed at stages T1 or T2 in 60.2% (101/168) of cases, with nodal involvement present in 66.6% (112/168) of cases at the time of primary diagnosis.

Median overall survival was 25 months. Locoregional recurrence occurred in 20.8% of patients with a median latency of 11 months.

### 3.2. PD-L1 Scoring and Overall Expression

To account for therapeutic relevance, PD-L1 positivity was defined as CPS ≥ 1 and TPS ≥ 1%, while CPS < 1 and TPS < 1% was defined as a negative score. CPS scores ranging from 1 to <20 were considered low, while scores ≥ 20 were considered high. For TPS, scores ranging from 1% to <50% were considered low, while scores ≥ 50% were considered high.

Overall, primary tumors were PD-L1 positive in 72% (121/168) of cases for CPS and 66.1% (111/168) for TPS, respectively. Untreated lymph node metastases were positive for CPS in 66.7% (36/54) and in 57.4% (31/54) for TPS. Distant metastases showed positivity for both CPS and TPS in 37.5% (3/8) of cases. Local recurrent tumors showed positivity for CPS and TPS in 59.3% (16/27) and 59.3% (16/27) of cases, respectively. Primary tumors showed high PD-L1 expression for CPS or TPS in 25.6% (43/168) and 6% (10/168) of cases, respectively. Untreated lymph node metastases were highly CPS-positive in 20.4% (11/54) of cases and highly TPS-positive in 5.6% (3/54) of cases. In distant metastases, 12.5% (1/8) of patients had high CPS and 0% (0/8) had high TPS. High CPS and TPS of local recurrent tumors occurred in 18.5% (5/27) and 7.4% (2/27) of cases (Figure 4).

### 3.3. PD-L1 Expression according to Tumor Locations

107 oropharyngeal squamous cell carcinomas, 39 oral cavity squamous cell carcinomas, 12 laryngeal squamous cell carcinomas, and 10 hypopharyngeal squamous cell carcinomas were compared for differences in PD-L1 expression.

PD-L1 expression was comparable between tumor sites for both CPS (*p* = 0.92) and TPS (*p* = 0.81) (Figure 5A,B). Expression of p16 in oropharyngeal cancers also had no effect on PD-L1 expression (CPS: *p* = 0.15; TPS: *p* = 0.36) (Figure 5C,D).

### 3.4. Intra-Tumoral Heterogeneity of Primary Tumors, Local Recurrent Tumors, and Metastases

Comparison of CPS between two intra-tumoral samples within 143 untreated primary tumors revealed statistically significant scoring differences (*p* = 0.0061). The same was true for 27 local recurrent tumors (*p* = 0.0289) (Figure 6A,C). In contrast, within 51 lymph node metastasis and 7 distant metastases, sample pairs did not differ significantly for absolute CPS (*p* = 0.243 and *p* = 0.469, respectively) (Figure 7A,C and Appendix A).

Absolute TPS values were significantly different when analyzing multiple samples for primary tumors (*p* = 0.017) and local recurrent tumors (*p* = 0.04) (Figure 6B,D). But lymph node metastases (*p* = 0.40) and distant metastases (*p* = 0.266) offered no significant differences in absolute TPS (Figure 7B,D and Appendix A).

To account for therapeutic relevance, intra-tumoral CPS scoring was also compared according to the following three categories: negative, i.e., CPS < 1; low, i.e., CPS 1 to <20; and high, i.e., CPS ≥ 20. When analyzing multiple samples, 28.7% (41/143) of primary untreated tumors were reclassified, and 22.2% (6/27) of recurrent tumors were reclassified. Importantly, 52.6% (27/57) of untreated primary tumor samples were reclassified from a negative score to a low score, and 10.5% (6/57) to a high score, respectively. For recurrent tumors, 27.3% (3/11) were reclassified from a negative score to a low score, and 0% (0/11) to a high score, respectively. Details can be found in Figure 8, Figure 9 and Appendix A.

Similarly, TPS-scoring was compared according to the categories negative, i.e., TPS < 1%; low, i.e., TPS 1% to <50%; and high, i.e., TPS ≥ 50% to account for therapeutic relevance. Here, primary untreated tumors were reclassified in 29.4% (42/143) of cases and 25.9% (7/27) of recurrent tumors. Importantly, 48.4% (30/62) of untreated primary tumor samples were reclassified from a negative score to a low score, and none were reclassified (0/62) to a high score, respectively. For recurrent tumors, 23.1% (3/13) were reclassified from a negative score to a low score, and 0% (0/13) to high score, respectively. Details can be found in Figure 8, Appendix A.

### 3.5. Scoring Concordance between Primary Tumors and Recurrences as Well as Primary Tumors and Their Metastases

#### 3.5.1. Primary Tumor vs. Local Recurrence

Samples of associated local recurrences were available for 13 primary tumors. Local recurrent tumors tended to show lower absolute CPS compared to the associated untreated primary tumor without achieving statistical significance (*p* = 0.19) (Figure 9A). Furthermore, in 61.5% (8/13) discordances occurred with tricategorical CPSs (Appendix A).

For TPS, absolute values were significantly lower in local recurrences (*p* = 0.01) (Figure 9B). There were also discordances of 50% (5/10) between the tricategorical TPSs (Appendix A).

#### 3.5.2. Primary Tumor vs. Associated Lymph Node Metastases and Distant Metastases

Inter-sample comparison of lymph node and distant metastases with their corresponding primary tumors showed no relevant differences for both absolute CPS (*p* = 0.57 and *p* = 0.63, respectively) and absolute TPS (*p* = 0.40 and *p* = 0.69, respectively) (Figure 10A–D and Appendix A). However, lymph node metastases differed from related primary tumors in 44.4% (16/36) of categorized CPS and in 41.6% (15/36) of categorized TPS.

### 3.6. Clinicopathological Correlations

#### 3.6.1. CPS and TPS Expression and Its Association with Recurrence and Metastases

Absolute CPS and TPS values of primary tumors were strongly associated with their distant metastases (CPS rho = 1; TPS rho = 1; *p* ≤ 0.01) and their lymph node metastases (CPS rho = 0.5, *p* ≤ 0.01; TPS rho = 0.5, *p* ≤ 0.01), but not with local recurrences (CPS rho = −0.21, *p* = 0.51; TPS rho = −0.14, *p* = 0.31) (Appendix A).

#### 3.6.2. CPS and TPS Expression and Its Association with Clinicopathological Features

Advanced UICC staging, tumor size, and positive lymph node involvement were weakly negatively correlated with binary (negative < 1; positive ≥ 1) CPS and TPS (Appendix A). Similar results were shown for tricategorical CPS (<1; ≥1<20; ≥20) and TPS (<1%; ≥1% to <50%; ≥50%).

CPS and TPS values were not correlated with clinical or pathological features such as tobacco or alcohol consumption, age, p16-status, grading, or extracapsular spread (Appendix A).

### 3.7. CPS and TPS Scores and Their Association with Patient Survival

In primary tumors, a CPS ≥ 1 and a TPS ≥ 1% were associated with a longer median overall survival of 102 months vs. 22.7 months (CPS ≥ 1 vs. <1; *p* = 0.005, HR: 0.46, 95%-CI: 0.27–0.78) and 102 months vs. 30.4 months (TPS ≥ 1% vs. <1%; *p* = 0.041, HR: 0.63, 95%-CI: 0.39–1.01) (Figure 11A,B).

For locally recurrent tumors, neither a CPS ≥ 1 nor a TPS ≥ 1% were associated with longer survival (median overall survival CPS ≥ 1 vs. <1: 11.4 months vs. 13.6 months, *p* = 0.42, HR: 1.4, 95%-CI: 0.61–3.29; median overall survival TPS ≥ 1% vs. <1%: 11 months vs. 13.2 months, *p* = 0.84, HR: 1.1, 95%-CI: 0.46–2.57) (Figure 11C,D).

CPS and TPS positivity in lymph node metastases was associated with superior 5-year overall survival (median overall survival CPS ≥ 1 vs. <1: 44.3 months vs. 20.4 months, *p* = 0.001, HR: 0.35, 95%-CI: 0.15–0.79; median overall survival TPS ≥ 1 vs. <1: 33.1 months vs. 21.4 months, *p* = 0.013, HR: 0.44, 95%CI: 0.21–0.93) (Figure 11E,F).

In multivariate Cox regression, a CPS ≥ 1 still demonstrated a reduced hazard ratio (Table 3). This was evident not only for the overall collective of HNSCC primary tumors, but also separately for tumor sites in the oropharynx and in the oral cavity.

For TPS, however, no relevant influence could be detected in the multivariate analysis as well as in the univariate analysis for oral cavity carcinomas. Only in the study population of HNSCC and OPSCC, a lower mortality rate was observed for a TPS ≥ 1 in univariate analysis (Table 4).

### 3.8. CPS and TPS Scores and Their Association with Response to Therapy

In 29 patients whose primary tumors were treated with radio-(chemo-)therapy, no correlations were found between treatment response of radio(chemo-)therapy and bicategorical CPS or TPS (CPS: contingency coefficient: 0.18, *p* = 0.61; TPS: contingency coefficient: 0.27, *p* = 0.29).

## 4. Discussion

The aim of this study was to analyze the degree and potential clinical impact of intra-tumoral PD-L1 expression heterogeneity. Further, we analyzed inter-tumoral heterogeneity between primary tumors and their metastases as well as recurrent tumors. Methodologically, we used standardized semiautomated staining technology to reduce technical bias. Subsequently, standardized evaluation according to the pharmDx HNSCC interpretation manual was performed using digital imaging software QuPath v3.4 to reduce further intra- and interrater bias [20]. In addition, representative tissue samples were selected after prior microscopic review.

First, we assessed absolute PD-L1 expression rates, and to which degree one primary tumor sample represents the PD-L1 expression of the entire tumor including its metastases.

Second, we addressed the question whether local recurrent tumors were immunologically similar to the primary tumor and whether we could demonstrate fluctuations in PD-L1 expression over time.

Third, we evaluated associations of PD-L1 expression with clinical and pathologic features.

The majority of HNSCC showed positivity for PD-L1 expression. Overall, 72% of all patients were positive for CPS, 66% for TPS in primary tumors, and 59.3% both for CPS and TPS in locally recurrent tumors, along with 66.7% and 57.4% in lymph node metastases of untreated tumors. However, the median CPS and TPS in primary tumors were only 4.2 and 3.8%, respectively. In 25.6% of primary tumors, CPS was high, i.e., CPS > 20.

Positivity rates were higher in our patient cohort compared to other reports where PD-L1 positivity for HNSCC patients has been shown to range from 21.6% to 64% [21,22,23,24]. Multiple factors may have contributed to our findings. While positivity rates did not statistically differ based on tumor localization in our patients, hypopharyngeal squamous cell carcinomas, which have been shown to present with lower PD-L1 expression [22], were somewhat underrepresented in our patient collective. Interestingly, p16-status did not affect PD-L1 expression in our patients while other authors have reported higher PD-L1 expression to be associated with p16 or HPV positivity, albeit in smaller sample sizes [25,26]. In particular, our findings may explain previous data on response to PD-L1-targeted immunotherapies in OPSCC, which showed no differences between HPV-positive and HPV-negative tumors. Only PD-L1 status had a significant impact on response [27]. Rates of positive PD-L1 expression of 72% in primary tumors and 55% in local recurrences are not consistent with objective response rates. However, the rates of high PD-L1 expression (CPS ≥ 20) of 25% in primary tumors and 17% in local recurrences in our cohort reflect relatively accurately the objective response rates reported to date in neoadjuvant or palliative studies [10,11].

From a methodological and technical perspective, comparability to other studies is somewhat compromised by differences in staining protocols, scoring algorithms, and cutoff values set for PD-L1 positivity. In addition, and perhaps most importantly, large discrepancies of PD-L1 expression ranging from 13.3% to 34% occur when comparing tissue from fully resected tumors to smaller biopsies. PD-L1 expression is typically lower in biopsies than in large tumor samples [28,29,30]. This is often the case because fewer stromal components are captured, and the highest number of immune cells is located at the tumor invasion front, perhaps contributing to the high degree of PD-L1 positivity in full tumor specimens versus biopsies. Therefore, the dominance of analyzed specimens compared to biopsies in this study may explain the higher PD-L1 positivity rates despite using TMAs.

### 4.1. Intra-Tumoral Heterogeneity

Previously reported concordance rates when comparing data from a single sample to repeated biopsy analyses from the same primary tumor ranged from 52% to 97.1% for CPS and 36% to 70% for TPS [14,24,29,31,32,33]. When assessing concordance rates based on therapy-relevant categories, discrepancy rates were notably lower but nevertheless substantial at up to 34% for CPS [31]. The discordance rates between categories have been shown to further increase, based on the number of samples analyzed within the tumor, reaching 64% in large-scale analyses [14]. Of note, the discrepancies more likely affect CPS than TPS, which is most likely due to a potential underrepresentation of the stroma in biopsies, as mentioned above [24,30,31].

In our patient collective, when comparing CPS and TPS from multiple samples obtained from a single, untreated primary tumor, we also found a substantial degree of discordance. Importantly, when comparing CPS and TPS based on categories relevant for therapy, i.e., negative, low, and high scores, obtaining multiple samples at the same time would have potentially resulted in a change of treatment regimen in 28.7% based on CPS and in 29.4% based on TPS of primary tumors if patients had received immunotherapy. Demonstrated intra-tumoral differences in PD-L1 expression may potentially explain the lack of complete response to neoadjuvant immunotherapies despite high CPS [10,11].

For local recurrent tumors, perhaps even more relevant for immunotherapeutic approaches according to current guidelines, the discordance rate was 18.5%. Based on a single sample, these patients would have been excluded from an immunotherapy-based treatment approach. Furthermore, in 13.4% of patients, a change from low CPS to high CPS (≥20) could have potentially avoided additional chemotherapy in favor of immunotherapy, which entails fewer side effects [6,12]. In this context, at the minimum, our data support a re-biopsy in cases of a negative CPS in primary or local recurrent tumors upon the initial analysis [34,35].

### 4.2. Discrepancies between Primary Tumors, Lymph Nodes, and Distant Metastases

When comparing absolute expression of CPS and TPS in primary tumors and associated lymph node metastases, statistically, there were no differences in our samples. However, after categorization of CPS and TPS into therapy-relevant categories, i.e., negative, low, and high, there were significant biological implications. Tumor microenvironment differed in 44.2% of patients if only one sample from the primary tumor or its associated metastases had been analyzed. TPS categories would have been affected in 41.6% of patients. Previous studies also show these potentially treatment-relevant differences in 15% to 44% of cases. Although not significant, in some cases pairwise comparisons between primary tumor and lymph node showed a substantially lower PD-L1 expression in the lymph node [28,32,36,37,38,39]. This may explain some of the differences in response between the primary tumor and lymph node observed in neoadjuvant immunotherapy seen in clinical trials [11,31,38,39,40].

There were even larger differences in the tumor microenvironment between primary tumors and distant metastases regarding PD-L1 expression. Absolute PD-L1 positivity was reduced in distant metastases compared to primary tumors (34.5% vs. 72.0%). This differs in particular from the previous description of increased PD-L1 expression with the presence of epithelial mesenchymal transition as the basis for metastasis [41]. Discordances of 14.3% between the primary tumor and associated distant metastasis also have therapeutic implications. In particular, the previously described deviations of 62% to 77% show clear biological differences, so that in the presence of non-resectable distant metastases, histological confirmation of this tissue appears to be useful for the treatment decision. Variation may be due to a small sample size, selection bias, or the antibody used [31,42]. 

### 4.3. Discrepancies between Primary Tumors and Recurrent Tumors

Although the direct comparison of PD-L1 expression based on CPS and TPS only showed differences in TPS between a primary tumor and local recurrence, the therapy-relevant categories of CPS and TPS differed. After classification of CPS and TPS into treatment-relevant categories, this study confirmed previously reported concordance of more than two thirds. For CPS, 77.8% and for TPS, 74.1% of cases were concordant [43,44]. However, this also means that there are potentially highly relevant differences of up to 26% in the treatment decision of patients. These differences appear to be even more important given the tendency for lower PD-L1 expression in local recurrences.

Currently, therapy choices for unresectable, non-irradiatable locally recurrent tumors are sometimes based on CPS analysis in a single sample of primary tumors, potentially including or excluding patients from immunotherapy who may profit from a different therapy regimen. In this context, our data support that even in the case of radiologically confirmed recurrence, at the minimum one re-biopsy should be performed to determine PD-L1 status prior to initiating therapy [34,35].

### 4.4. PD-L1 Expression and Survival

In our collective, high CPS was associated with a favorable prognosis in all primary tumors, i.e., HNSCC (HR = 0.46), OSCC (HR = 0.23), OPSCC (HR = 0.46), and associated lymph node metastases (HR = 0.1) and, to a lesser extent, in local recurrences (HR = 0.37; 95%-CI: 0.11–1.28). While our findings were consistent across tumor types, previous study results are heterogeneous regarding the prognostic value of PD-L1 expression [25,39,45,46,47,48]. Some studies even showed a negative association between PD-L1 expression and patients’ prognosis [25,39,46].

As mentioned above, overall PD-L1 levels measured in our patients were relatively high compared to other studies, most likely attributable to technical aspects and sample-collection methodology. The latter, which included the analysis of whole tumor samples rather than biopsies may have helped to eliminate a potential bias since others previously suggested a positive correlation between PD-L1 expression and tumor size, which was not evident in our patient collective [49].

Compared to TPS, CPS had higher prognostic impact in our patients. These findings support the data of others who demonstrated the prognostic benefit of immune cell infiltration in tumor tissue [20,44,47,50,51]. As pointed out by Fukushima et al., a high level of immune cell infiltration in p16-negative OPSCC leads to patient survival rates comparable to p16 positive tumors with low immune cell infiltration rates [27]. In addition to T-cell infiltration density, T-cells with a conserved IFN-g signature are associated with a good clinical response to immune checkpoint therapy and a favorable outcome. IFN-g, as a key inducer of PD-L1, may most likely explain this phenomenon [52,53]. One other explanation previously explored by our group (Kürten et al., 2021) would be that PD-L1 is largely expressed on macrophages in the head and neck tumor microenvironment, a fact that is captured by the CPS, but not the TPS [54]. 

Both CPS and TPS were not associated with patients’ response to radiochemotherapy. Rather, infiltration of CD8-positive T-cells and PD-L1 expression on immune cells are shown to be predictive biomarkers of pathologic response prior to combined induction chemoimmunotherapy and following radioimmunotherapy [55]. However, chemo- and radiotherapy-induced neoantigen liberation prior to adjuvant immunotherapy seems to be a decisive factor [56]. On the other hand, chemotherapy and radiotherapy may also potentially compromise the function of local tumor-resident and peripheral blood-localized T-cells needed for immunotherapy. Here, further research is required though, since the analyzed sample size of patients undergoing primary radiotherapy was low.

### 4.5. Limitations

Clinical data and tissue collection were performed retrospectively, which may have resulted in selection bias due to differences in treatment, tumor stage, and gender within the study population. Furthermore, missing tumor tissue samples and patient files during data collection led to selection and information bias. Some missing survival data of this study population led to the loss of follow-up cases in the survival analysis. Additionally, inter-sample comparison of primary tumors with corresponding distant metastases is possibly affected, due the low sample size. From a methodological and technical perspective, comparability with other studies is somewhat compromised by differences in staining protocols, scoring algorithms, and cutoff values set for PD-L1 positivity. In addition, discrepancies in PD-L1 expression may be influenced by the analysis of tissue microarrays despite representative cores taken from the tumor stroma border out of the whole specimen. This could lead to lower PD-L1 expression in tissue microarrays due to fewer stromal components, even though this collection shows high PD-L1 expression, suggesting representative tissue cores. These points need to be considered when discussing the results of this study.

## 5. Conclusions

Our results reveal a substantial degree of clinically relevant intra- and inter-sample heterogeneity in PD-L1 expression stained by standardized, recommended automated staining protocols and measured by digital image analysis of CPS and TPS. Intra-tumoral heterogeneity of PD-L1 expression is particularly present in primary and locally recurrent tumors, but not in metastases. In contrast, no statistically relevant inter-tumoral differences were observed between primary tumors and their metastases, but lower PD-L1 expression was observed in locally recurrent tumors. Primary tumor location and p16 expression in OPSCC did not affect PD-L1 expression. However, inter- and intra-tumoral heterogeneities between samples were particularly evident after classification into the therapy-relevant categories of CPS and TPS. Therefore, multiple tissue samples should be tested for PD-L1 expression to account for tissue heterogeneity. Especially, local recurrent tumors should be re-biopsied and re-assessed for CPS prior to treatment initiation. For patients who are candidates for immunotherapy, obtaining multiple tumor samples appears to be necessary to adequately determine their respective levels of PD-L1 expression and determine their eligibility for targeted immunotherapy. Increased PD-L1 expression in primary tumors and lymph node metastases is associated with improved overall survival. Patients with negative PD-L1 expression in primary tumors or lymph node metastases have a worse prognosis and may require intensified treatment and observation.

## Figures and Tables

**Figure 1 cancers-16-02103-f001:**
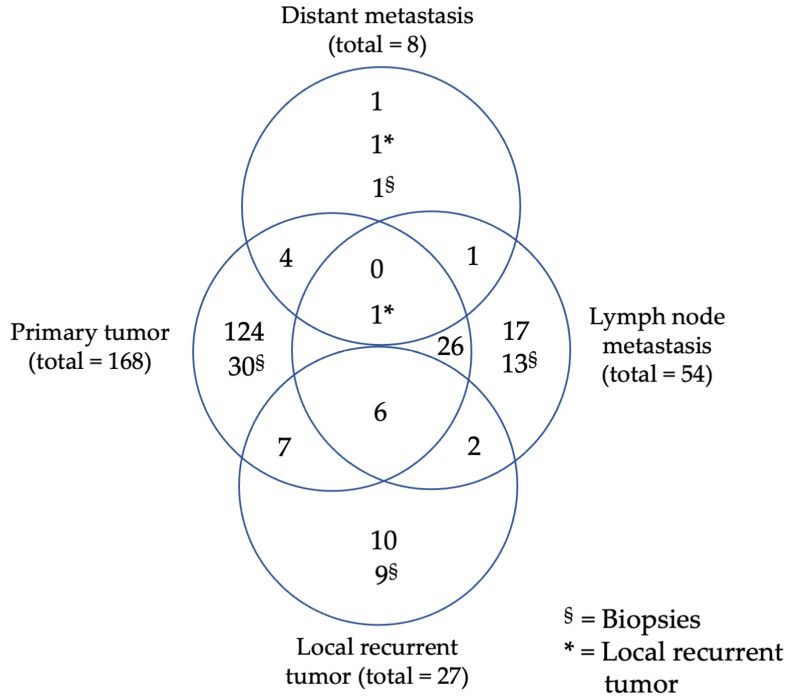
Study population, Venn diagram of specimen distribution out of primary tumors, local recurrent tumors, lymph node or distant metastasis.

**Figure 2 cancers-16-02103-f002:**
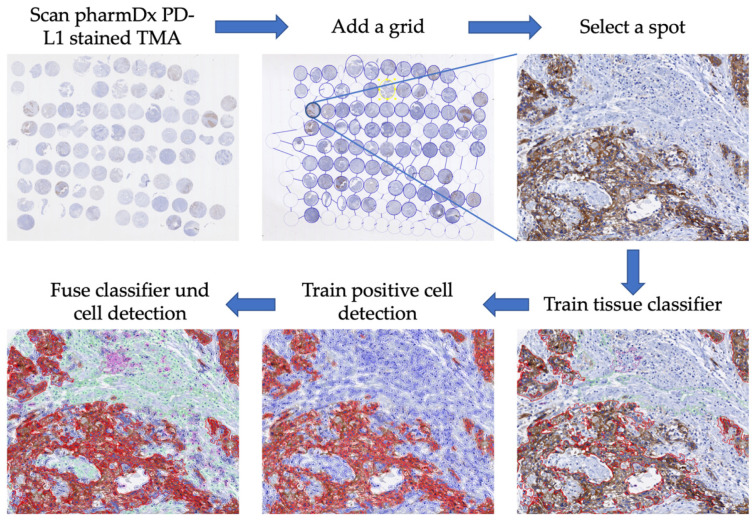
Procedure for the automated analysis of PD-L1 expression using QuPath v3.4; 1. Scanning of immunohistochemical tissue micro array PD-L1 22C3 pharmDx staining; 2. Importing picture into QuPath and adding a tissue micro array grid; 3. Selecting and focusing on spots; 4. Training tissue classifier (immune cells, stroma, tumor); 5. Training positive cell detection; 6. Fusing results of tissue classifier and positive cell detecting.

**Figure 3 cancers-16-02103-f003:**
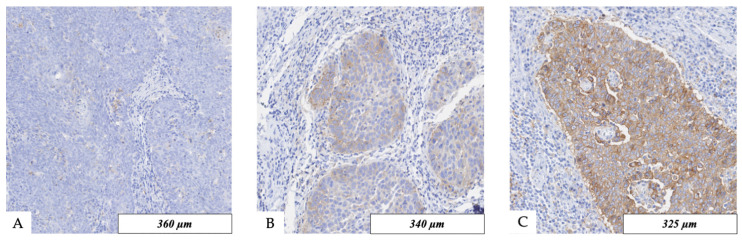
(**A**–**C**) Immunohistochemical PD-L1 staining with 22C3 pharmDx anti-human PD-L1 antibody; (**A**) no PD-L1 expression; (**B**) moderate intra-tumoral PD-L1 expression; (**C**) strong intra-tumoral PD-L1 expression.

**Figure 4 cancers-16-02103-f004:**
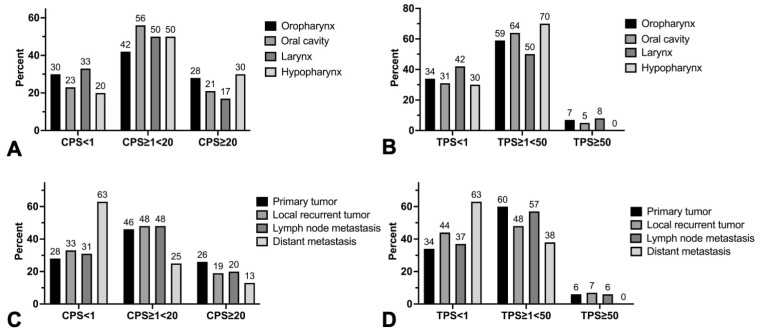
Frequencies of CPS categories in head and neck squamous cell primary tumors and metastasis of different locations (**A**–**D**) Frequency distributions of tricategorical combined positive score (CPS) (<1, ≥1<20, ≥20) and tumor proportion score (TPS) (<1, ≥1<50, ≥50) in percent for (**A**,**B**) tumor sites in head and neck squamous cell cancer and (**C**,**D**) different kinds of tumors (primary tumor/local recurrent tumor/lymph node or distant metastasis). No statistically significant differences were found in frequencies of CPS and TPS categories among different tumor sites and kind of tumors.

**Figure 5 cancers-16-02103-f005:**
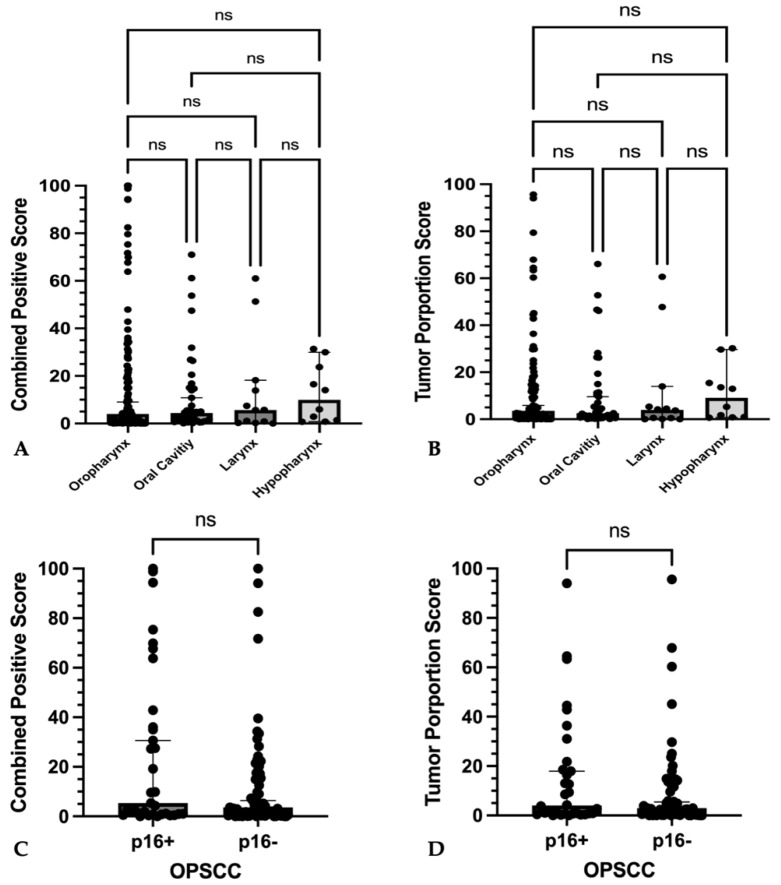
PD-L1 expression among different locations and differences to p16-expression (**A**–**D**) Graphs show single plotted values with median and 95% confidence interval as whiskers. No significant differences in (**A**) combined positive score or (**B**) tumor proportions score between primary tumors of different location could be found. No significant differences of (**C**) combined positive score and (**D**) tumor proportion score between p16-positive or p16-negative oropharyngeal squamous cell carcinoma could be found. No statistically significant differences (ns) were found among absolute CPS and TPS values as well as p16-positive and p16-negative OPSCC.

**Figure 6 cancers-16-02103-f006:**
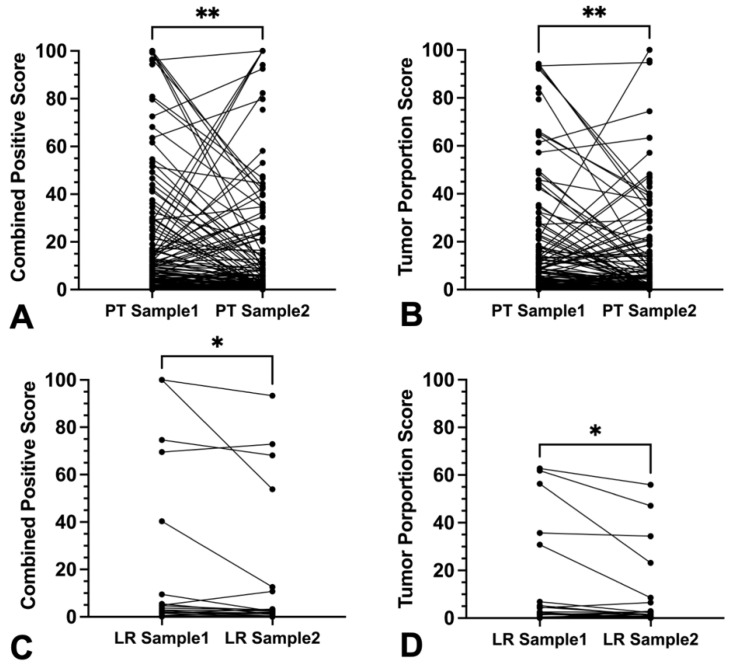
Discordance of CPS and TPS in separate punches from the same primary tumor or local recurrent tumor specimen. (**A**,**B**) Plotted values of *PD-L1* expression in primary tumor (PT) tissue samples and of (**C**,**D**) local recurrent tumors (LR) (**A**,**B**) showed significant differences of combined positive score and tumor proportions score ((**A**) ** *p* < 0.01; (**B**) ** *p* ≤ 0.01); (**C**,**D**) showed significant differences of combined positive score and tumor proportions score ((**C**) * *p* < 0.05; (**D**) * *p* ≤ 0.05).

**Figure 7 cancers-16-02103-f007:**
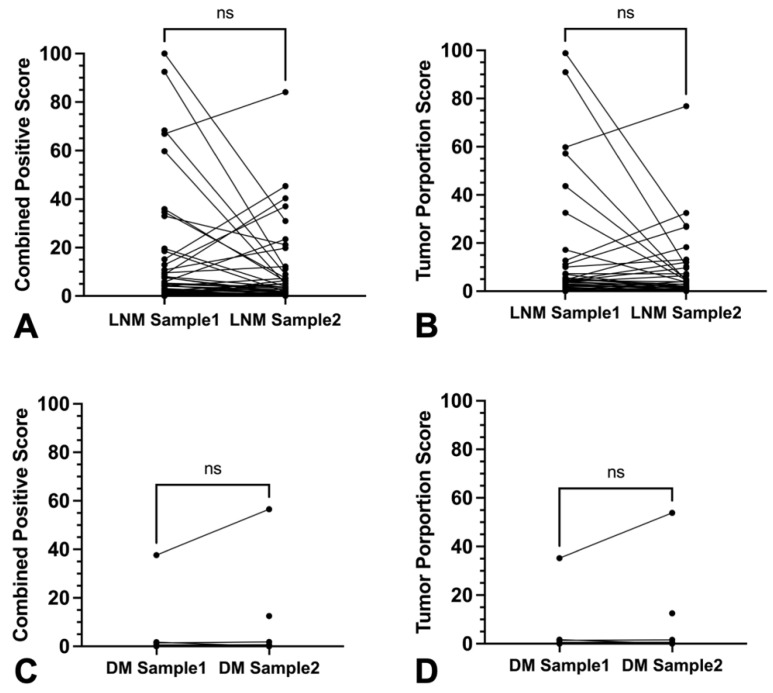
Discordance of CPS and TPS in separate punches from the same lymph node or distant metastases specimen (**A**–**D**). (**A**,**B**) Plotted values of PD-L1 expression in lymph node metastases (LNM) tissue samples and of (**C**,**D**) distant metastases (DM). (**A**–**D**) showed no significant differences of combined positive score and tumor proportions score ((**A**–**D**) ns *p* > 0.05).

**Figure 8 cancers-16-02103-f008:**
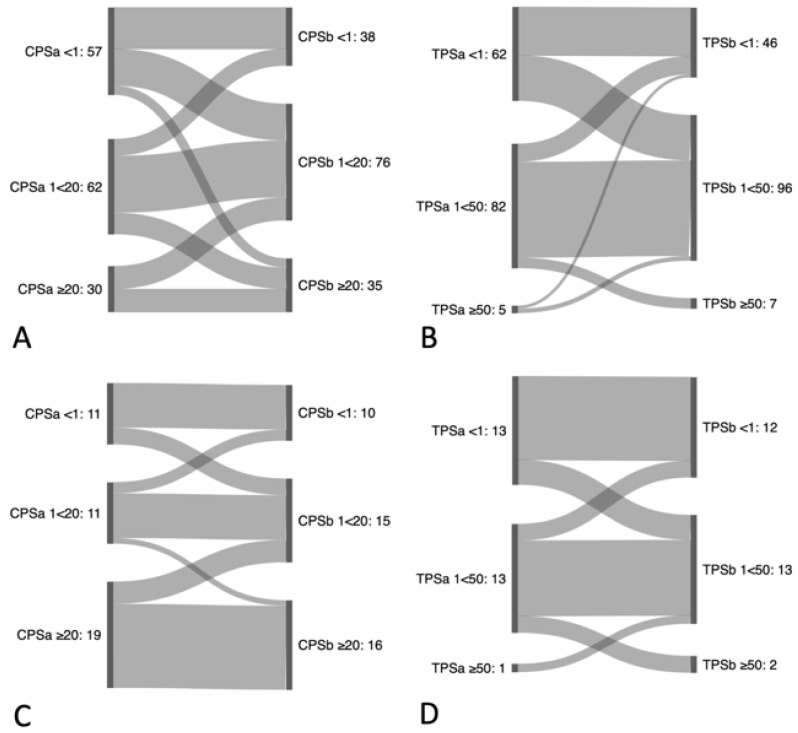
Reclassification of CPS and TPS based on analysis of multiple samples. Sankey diagram of (**A**) combined positive score (CPS) categories (<1; ≥1<20; ≥20) and (**B**) of tumor proportion score (TPS) categories (<1%; ≥1%<50%; ≥50%) between two different intra-tumoral biopsies of primary tumors, as well as (**C**) of CPS (<1; ≥1<20; ≥20) and (**D**) of TPS (<1%; ≥1%<50%; ≥50%) between two different intra-tumoral biopsies of local recurrent tumors.

**Figure 9 cancers-16-02103-f009:**
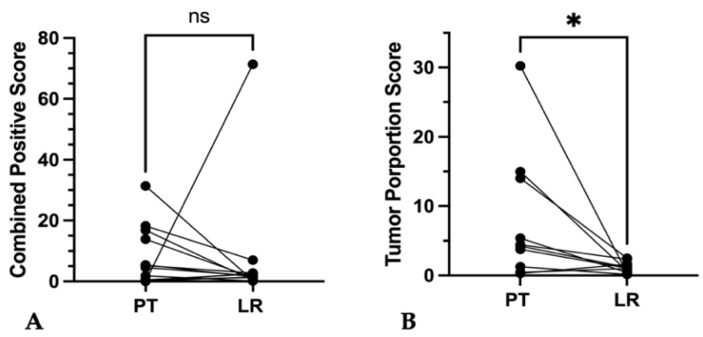
Inter-tumoral discordance of CPS and TPS between primary tumor and local recurrent tumor of the same patients. (**A**,**B**) Graphs show changes in (**A**) combined positive score (CPS) (ns *p* > 0.05) and (**B**) tumor proportion score (TPS) (* *p* = 0.01) of local recurrent tumors (LRs) compared to paired primary tumors (PTs).

**Figure 10 cancers-16-02103-f010:**
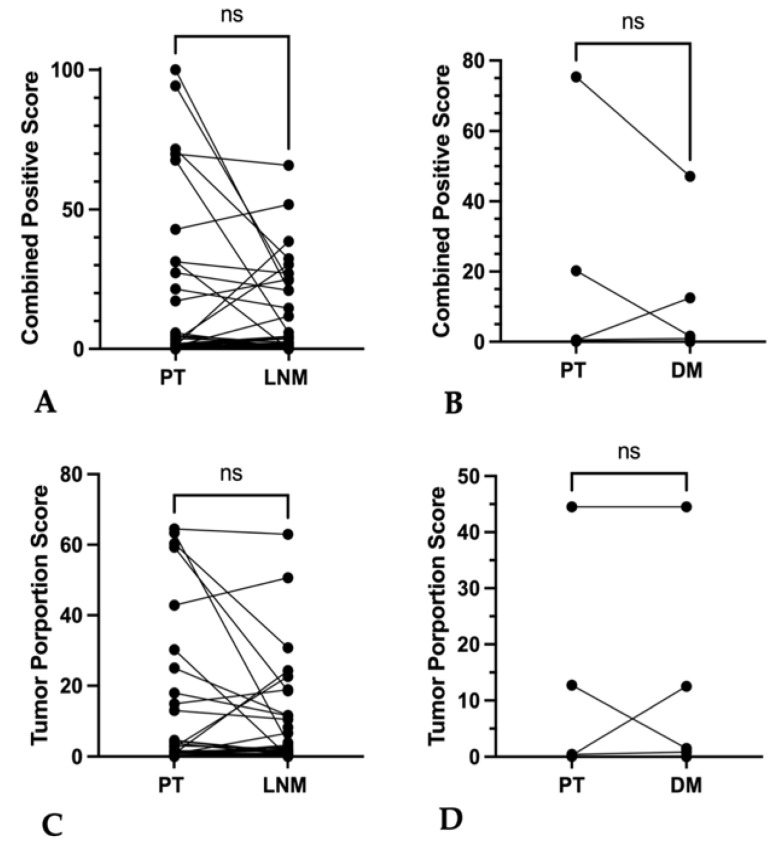
Inter-tumoral discordance of CPS and TPS between primary tumor and lymph node or distant metastases of the same patients. (**A**,**B**) Graphs show CPS in paired specimen of (**A**) primary tumor (PT) and lymph node metastases (LNMs); (**B**) primary tumor (PT) and distant metastases (DMs) without statistically significant differences (ns *p* > 0.05); (**C**,**D**) Graphs show TPS in paired specimen of (**C**) primary tumor (PT) and lymph node metastases (LNMs); (**D**) primary tumor (PT) and distant metastases (DMs) without statistically significant differences (ns *p* > 0.05).

**Figure 11 cancers-16-02103-f011:**
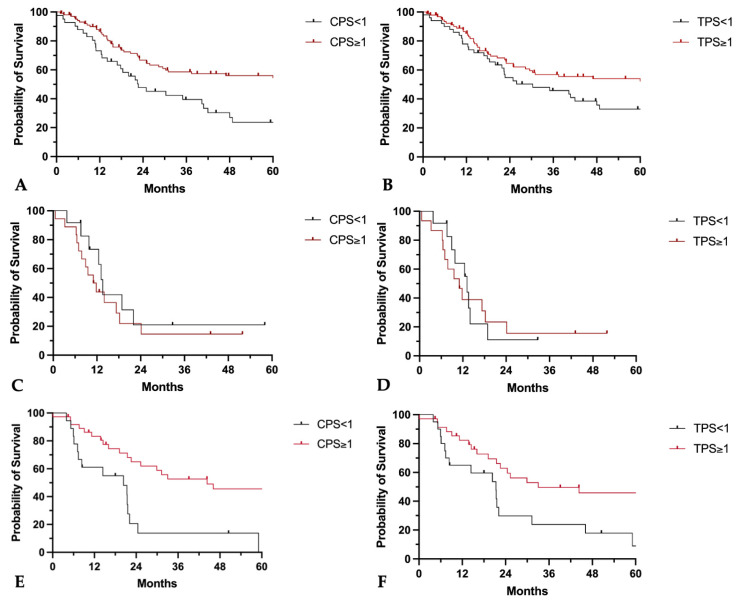
Prognostic implications of CPS and TPS on patients’ overall survival. (**A**–**D**) Kaplan–Meier curves of five-year overall survival probability in (**A**,**B**) primary tumors, (**C**,**D**) local recurrent tumors, and (**E**,**F**) lymph node metastases separated for (**A**,**C**,**E**) CPS < 1 or ≥1 and for (**B**,**D**,**F**) TPS < 1 or ≥1.

**Table 1 cancers-16-02103-t001:** Study population, origin of specimen, total number, and percentages of specimens and patients.

Tissue Samples	Specimen No. (%)	Patients No. (%)
primary tumors	198 (63.9)	168 (65.4)
local recurrent tumors	36 (11.6)	27 (10.5)
lymph node metastasis	67 (21.6)	54 (21.0)
distant metastasis	9 (2.9)	8 (3.1)

**Table 2 cancers-16-02103-t002:** Clinicopathological data of patients with primary tumors.

**Sex**	**No. (%)**
female	53 (31.5)
male	115 (68.5)
**Age in years median (range)**	
female	62 (39–80)
male	60 (19–80)
**Body mass index**	
median (range)	24.9 (13.1–43)
**Overall survival in months**	
median (range)	25 (1–137)
**Primary tumor site**	**No. (%)**
oropharynx	107 (63.7)
p16 positive	35 (32.7)
p16 negative	72 (67.3)
oral cavitiy	39 (23.2)
larynx	12 (7.1)
hypopharynx	10 (6.0)
**T (UICC 7. edition)**	**No. (%)**
1	51 (30.4)
2	52 (31.0)
3	34 (20.2)
4	31 (18.5)
**N (UICC 7. edition)**	**No. (%)**
0	59 (35.1)
1	31 (18.5)
2a	8 (4.8)
2b	42 (25.0)
2c	26 (15.5)
3	2 (1.2)
**Stage (UICC 7. edition)**	**No. (%)**
I	20 (11.9)
II	24 (14.3)
III	35 (20.8)
IVa	68 (40.5)
IVb	3 (1.8)
IVc	18 (10.7)
**Grading**	
I	6 (3.6)
II	89 (53.0)
III	52 (31.0)
**Lymphnode ratio**	
median (range) with ≥ 15 resected lymph nodes	0.08 (0–0.6)
**Extranodal extension**	
Yes	30 (17.3)
**Toxicant use**	**No. (%)**
smoking	119 (70.8)
alcohol abuse	58 (34.5)
**Therapy**	
surgery	24 (14.3)
surgery ± postoperative radio(chemo-)therapy	47 (28.0)
radio(chemo-)therapy	29 (17.3)

**Table 3 cancers-16-02103-t003:** Hazard ratios for CPS as prognostic biomarker for overall survival in HNSCC and different subsites, results of univariate and multivariate Cox regression for CPS ≥ 1 in primary tumors with clinical and pathological features.

Tumor Site	Covariates	N	Hazard Ratio	95% Confidence Intervall	*p*-Value
HNSCC	univariate	149	0.4553	0.2885 to 0.7261	0.0008
	staging	147	0.5385	0.3372 to 0.8693	0.01
OPSCC	univariate	107	0.4418	0.2422 to 0.8126	0.008
	staging, p16	107	0.4561	0.2419 to 0.8708	0.016
OSCC	univariate	32	0.2328	0.08298 to 0.6685	0.005
	staging	32	0.2344	0.08326 to 0.6751	0.006

**Table 4 cancers-16-02103-t004:** Hazard ratios for TPS as prognostic biomarker for overall survival in HNSCC and different subsites, results of univariate and multivariate Cox regression for TPS ≥ 1 in primary tumors with clinical and pathological features.

Tumor Site	Covariates	N	Hazard Ratio	95% Confidence Intervall	*p*-Value
HNSCC	univariate	149	0.6192	0.3938 to 0.9789	0.04
	staging	147	0.689	0.4342 to 1.100	0.12
OPSCC	univariate	96	0.5373	0.2944 to 0.9834	0.04
	staging, p16	92	0.4957	0.2629 to 0.9399	0.03
OSCC	univariate	32	0.3885	0.1422 to 1.061	0.06
	staging	32	0.5015	0.1756 to 1.432	0.19

## Data Availability

Raw data can be requested on demand by mail to the corresponding author.

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
