# Peer review of "Standardized Digital Image Analysis of PD-L1 Expression in Head and Neck Squamous Cell Carcinoma Reveals Intra- and Inter-Sample Heterogeneity with Therapeutic Implications"

_cancers, 2024, doi:10.3390/cancers16112103_

Round 1
Reviewer 1 Report
Comments and Suggestions for Authors
The article represents a digital distal analysis of the expression of PD-1 (mainly) in head and neck cancer and its therapeutic implications. The article has a clear rationale, the methods are well described, and the analysis of the samples was performed adequately. The reports highlight several important issues in the clinic and clinical design. I would suggest the authors summarize the limitations that the correctly point out in the discussion in a section at the end of the manuscript because I think it will be easily viewed by the reader.
Author Response
Thank you for your comments, we feel that our revisions according to your suggestions have significantly improved the manuscript. We added a separate section at the end to highlight possible bias and limitations.

Reviewer 2 Report
Comments and Suggestions for Authors
This study has potential impact for cancer treatment.
Comments:
1. Please list all abbreviations.
2. Please add details of antibody use including working dilution in Materials and Methods area.
3. Please add scale bar in Figure 3 images.
4. On Table 2: please add patients' BMI. What is the median age of male and female patients? What is the relationship between p16 staining and PD-L1 in this study? Any patients use(s) betel nuts?
5. On Figure 4: any statistical analysis available?
6. On Figure 5: it will be better to add "N" on each group on the graph.
6. Pease discuss the limitation(s) of the current study?
Author Response
Dear SIr or Madame,
we would like to thank you very much for your time and effort in improving the content of the manuscript by your comments. We feel that our revisions according to your suggestions have significantly improved the manuscript.
- Please list all abbreviations.
You find a summary of all abbreviations at the end of the manuscript.
- Please add details of antibody use including working dilution in Materials and Methods area.
We added more details on staining protocol in the material and methods section.
PD-L1 IHC 22C3 pharmDx kit of Dako (Agilent Technologies, Santa Clara, CA, United States) is a ready to use solution without further need of dilution, so that we cannot offer further information, unfortunately.
- Please add scale bar in Figure 3 images.
Thanks for your comment, scale bars have been added for better orientation and comparison.
- On Table 2: please add patients' BMI. What is the median age of male and female patients? What is the relationship between p16 staining and PD-L1 in this study? Any patients use(s) betel nuts?
Clinical informations were added in Table 2. The patient cohort is from middle Europe, so that betel nut and coca leaves were not used regularly. We could not demonstrate any correlation or differences in PD-L1 expression between p16-positive and negative groups. Results were shortly described in sections 3.3 and 3.6.2 with further information attached in figure 5 and the supplement S3-4.
- On Figure 4: any statistical analysis available?
P Values of Fishers Exact test or Chi Square test were mentioned in section 3.3.
- On Figure 5: it will be better to add "N" on each group on the graph.
Thank you for your suggestion to make the illustration clearer. However, to avoid misunderstandings, we have decided to continue to use ns usually used as not significant, as N is already used for the number of patients.
- Pease discuss the limitation(s) of the current study?
We added a separate section at the end to highlight possible bias and limitations.
Sincerly yours
Eric Deuss

Reviewer 3 Report
Comments and Suggestions for Authors
The study, titled "Standardized Digital Image Analysis of PD-L1 Expression in Head and Neck Squamous Cell Carcinoma Reveals Intra- and Inter-Sample Heterogeneity with Therapeutic Implications," examines the variability of PD-L1 expression in different tumor samples from individual patients with head and neck squamous cell carcinoma (HNSCC). This variability has significant implications for treatment, particularly in the context of immunotherapy.
Through the use of multi-section staining techniques, the study reveals significant changes in the Combined Positive Score (CPS), an important measure for assessing PD-L1 expression. The results suggest that the use of a single tumor sample for PD-L1 assessment could lead to an underestimation of the marker’s expression, potentially impacting treatment decisions. In particular, intra-tumor heterogeneity was found to influence treatment decisions in 28.7% of cases. This percentage increases to 44.4% when primary tumors are compared with their metastatic counterparts, and even to 61.5% when first tumors are compared with local recurrences.
In addition, the study found that increased CPS levels in primary tumors and lymph node metastases were correlated with improved 5-year overall survival. These results underscore the need to evaluate multiple sections from different tumor sites in HNSCC patients, especially when initial tests for PD-L1 are negative. Such a comprehensive evaluation ensures a more accurate assessment of PD-L1 expression and leads to more informed decisions about eligibility for immunotherapy.
This study underscores the critical need for standardized, multiregional analysis of PD-L1 to account for variability in expression and thereby improve the precision of therapeutic approaches for patients with squamous cell carcinoma of the head and neck.
What are the key findings of the study on PD-L1 expression in squamous cell carcinoma of the head and neck?
How might the results of this study impact therapeutic approaches for patients with this type of cancer?
What is the significance of the heterogeneity of PD-L1 expression within and between samples for personalized medicine in cancer treatment?
Author Response
Dear Sir or Madame,
Thank you for your comments, we feel that our revisions according to your suggestions have significantly improved the manuscript. Espacially in highlighting key findings and relevant implications for treatment. We have revised the summary in response to your suggestions and have included the answers to your questions.
What are the key findings of the study on PD-L1 expression in squamous cell carcinoma of the head and neck?
- There is relevant intratumoral heterogeneity of PD-L1 expression in primary and locally recurrent tumors, but not in metastases.
- There are no statistically relevant intertumoral differences between primary tumors and their metastases, but lower PD-L1 expression in local recurrent tumors.
- There is a relevant number of therapy relevant changes in CPS and TPS categories.
- Primary tumor location and p16 expression in OPSCC did not affect PD-L1 expression.
- Increased PD-L1 expression in primary tumors and lymph node metastases is associated with improved overall survival.
How might the results of this study impact therapeutic approaches for patients with this type of cancer?
- Multiple tissue specimen should be analyzed especially in case of negative PD-L1 Scores CPS and TPS.
- Patients with negative PD-L1 expression in primary tumors or lymph node metastasis show worse prognosis and maybe need intensified treatment and observation.
What is the significance of the heterogeneity of PD-L1 expression within and between samples for personalized medicine in cancer treatment?
- Multiple samples should be analyzed for PD-L1 expression scores such as CPS to account for tumor heterogeneity and, if possible, to better predict the expected treatment response.
- In case of PD-L1 negativity at least one additional section should be analyzed in order to avoid exclusion of immunotherapy.
- Local recurrent tumors should be re-biopsied and new CPS determination should be initiated prior to treatment
Sincerly yours
Eric Deuss

Round 2
Reviewer 2 Report
Comments and Suggestions for Authors
No more comments